# Implementation of Fire Policies in Brazil: An Assessment of Fire Dynamics in Brazilian Savanna

**Ananda Santa Rosa de Andrade** [1,*] ⓘ, **Rossano Marchetti Ramos** [2], **Edson Eyji Sano** [3] ⓘ, **Renata Libonati** [4,5] ⓘ, **Filippe Lemos Maia Santos** [4,6] ⓘ, **Julia Abrantes Rodrigues** [4] ⓘ, **Marcos Giongo** [7], **Rafael Rodrigues da Franca** [1] and **Ruth Elias de Paula Laranja** [1]

1   Department of Geography, University of Brasilia, Brasília 70910-900, Brazil; rrfranca@unb.br (R.R.d.F.); laranja@unb.br (R.E.d.P.L.)
2   National Center for the Prevention and Fighting of Forest Fires, Brazilian Institute of the Environment and Renewable Natural Resources, Brasília 70818-900, Brazil; rossano.ramos@ibama.gov.br
3   Brazilian Agricultural Research Corporation, Planaltina 73301-970, Brazil; edson.sano@embrapa.br
4   Department of Meteorology, Federal University of Rio de Janeiro, Rio de Janeiro 21941-919, Brazil; renata.libonati@igeo.ufrj.br (R.L.); filmaias@hotmail.com (F.L.M.S.); abrant.julia@gmail.com (J.A.R.)
5   Center for Forestry Studies, Higher Institute of Agronomy, University of Lisbon, 1349-017 Lisbon, Portugal
6   Graduate Program in Climate and Environment (CLIAMB), National Institute for Research in the Amazon (INPA)/State University of Amazonas (UEA), Manaus 69069-001, Brazil
7   Center for Environmental Monitoring and Fire Management, Department of Forest Engineering, Federal University of Tocantins, Tocantins 77402-970, Brazil; giongo@uft.edu.br
*   Correspondence: anandasrandrade@gmail.com

**Abstract:** In 2012, the Brazilian government implemented the Federal Brigades Program (FBP), a fire policy strategy to hire and train firefighters to combat wildfires. This study analyzed the impact of this program on fire behavior before (2008–2012) and after (2013–2017) its implementation in the Parque do Araguaia Indigenous Land, the largest indigenous territory with the highest occurrence of fires in the Brazilian tropical savanna. We analyzed the annual pattern of fire incidence in the dry season, the fire impact per vegetation type, the recurrence, and the relationship between fire and precipitation. The datasets were based on active fire products derived from the Moderate Resolution Imaging Spectroradiometer (MODIS), the Landsat and Resourcesat-based burned area products, and the records of the fire combat operations. Our results showed that FBP contributed to the reduction of the number of areas affected by fires and to the formation of a more heterogeneous environment composed of fire-resistant and fire-sensitive native vegetation fragments. On the other hand, after the implementation of the FBP, there was an increase in the recurrence of 3–4 years of fires. We concluded that the FBP is an important public policy capable of providing improvements in fire management activities.

**Keywords:** fire pattern; firefighters; indigenous land; environmental policy

## 1. Introduction

Wildfires disturb natural environments, affect socio-economic activities, and put the health and the life of people in danger [1]. More specifically, wildfires consume most of the surface organic layers and reduce stream water quality by increasing the inputs of soil erosional sediments and fertilizers [2]. Fires cause landscape fragmentation [3], loss of biodiversity [4], and release carbon stored in biomass and soils into the atmosphere, mostly in the form of $CO_2$, but also as $CO$, $CH_4$, and $CH_3Cl$ [5–7]. Whenever the dry season is prolonged, fire severity and intensity can be aggravated due to the increase in the dry organic fuels that can cause extensive and uncontrolled wildfires [8–10]. Moritz and Stephens [11] recommended that future cities and rural communities should be created considering the low susceptibility to wildfires.

In Brazil, fire in rural areas is mainly associated with illegal land cleaning for land speculation, land preparation for crop cultivation, pasture renewal, and charcoal production [12]. Fire prevention policy in Brazil is mainly based on field environmental law enforcement operations to combat and control illegal fire activities [13,14]. In addition, to reduce the negative consequences of fires and to comply with the goals established in international agreements, Brazil has regulated rules on the use of fire [15], created specific databases and public policies to support fire monitoring activities [16,17], and fostered projects based on technical cooperation agreements with other countries [18,19]. The work of the firefighters is also another important means to carry out fire prevention activities, environmental education, and combat wildfires [20,21]. The Federal Brigades Program (FBP) is part of the Sectorial Health Plan for the Mitigation and Adaption to Climate Change [22]. The objective of the FBP is to train and hire firefighters to reduce the extent and the number of fire occurrences during the dry season, combat wildfires, and carry out activities to prevent uncontrolled fires [23]. Brigades are hired whenever the Ministry of Environment (MMA) declares year-based emergency periods during the dry seasons in different regions of Brazil. The FBP operates continuously, with an average cost of ~$4.5 \times 10^6$ U.S. dollars in the period 2013–2017 since 2012 [23]. This is a rather low budget if compared with those from other countries, for example, $1.5 \times 10^9$ U.S. dollars in the period 2010–2020 in the United States [24] or $870 \times 10^9$ AU dollars in 2019/2020 in Australia [25].

Most of the scientific publications related to the rural fire brigades deal with safety, health, and psychological issues [26–28]. There are few studies investigating the effectiveness of rural fire brigades hired to reduce burned areas. An exception is a recent research study conducted by Oliveira et al. [29], which reported a reduction of 12%, on average, of burned areas in conservation units located in the Brazilian Cerrado (tropical savanna) with brigades, compared with those units without brigades. The same study found a reduction of an additional 6% of burned areas in the Cerrado´s conservation units with both fire suppression and prevention activities in comparison with those units with only suppression practices.

After decades of frustrating attempts of implementing zero-fire policies, Brazil started to implement fire management policies, reintroducing fire as a management tool in fire-prone ecosystems, which is the case of the Brazilian Cerrado [30,31]. Firefighters hired by the FBP are important players of these policies. They can act not only in federal conservation units of integral protection or sustainable use but also in indigenous lands, public forests, federal rural settlements, and *quilombos*. The research question of this study is: how effective is the FBP in reducing the size, number, and recurrence of fires in the Brazilian Cerrado? To our best knowledge, there is a lack of studies regarding the effectiveness of the FBP-based actions to prevent wildfires in Brazil. To evaluate these possible effects of FBP, this study aimed to compare the spatial variability of fires before (2008–2012) and during (2013–2017) the implementation of the FBP in the Parque do Araguaia Indigenous Land.

## 2. Materials and Methods

### 2.1. Study Area

The study area corresponds to the Parque do Araguaia Indigenous Land (Figure 1), with an approximate surface area of $13 \times 10^3$ km$^2$. This territory corresponds to a large seasonal wetland and the intracratonic Quaternary sedimentary basin of South America, flooded mainly by rainfall and groundwater [32,33]. The annual flooding occurs during the rainy season mainly from the Javaés and Araguaia rivers [32]. The region is a mosaic of phytophysiognomies that are either resistant to fire (shrublands and wooded grasslands) or sensitive to fire (forestlands) [34]. The climate is tropical, has an average annual rainfall ranging from 1200 mm to 1800 mm [34], and an average temperature ranging from 22 °C to 26 °C [35]. The dry season occurs between May and October, while the rainy season occurs between October and April.

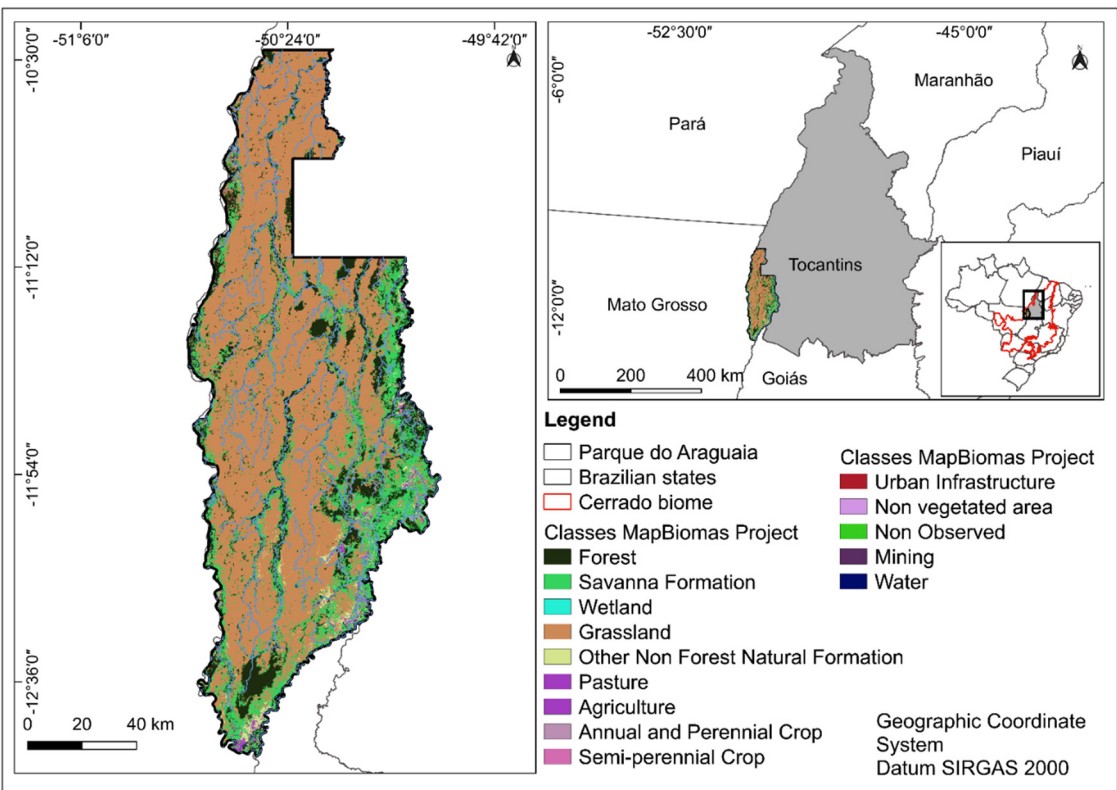

**Figure 1.** Location of the Parque do Araguaia Indigenous Land in the Tocantins State and in the Cerrado biome and the main land cover classes according to MapBiomas. Source: Ramsar [36].

Fires in indigenous lands are related to traditional cultivation methods [37] or illegally set by hunters and land grabbers [38]. Since 2013, the Parque do Araguaia Indigenous Land is one of the IBAMA's highest priority areas to mitigate degradation caused by wildfires, with 217 contracts awarded between 2013 and 2017. Most of the hired firefighters were indigenous people residing in this Park (Table 1).

**Table 1.** Total number of firefighters hired by the Federal Brigade Program in the Parque do Araguaia Indigenous Land in the period 2013–2017.

| Year | Number of Hired Firefighters |
|------|------------------------------|
| 2013 | 15 |
| 2014 | 44 |
| 2015 | 58 |
| 2016 | 50 |
| 2017 | 50 |
| Total | 217 |

*2.2. Datasets*

We selected the Moderate Resolution Imaging Spectroradiometer (MODIS) Active Fire product (MCD14ML), collection 6, which is derived applying thresholds to the brightness temperatures from the middle-infrared and thermal infrared spectral bands of the MODIS sensor onboard the Aqua and Terra platforms (https://firms.modaps.eosdis.nasa.gov/download/, accessed on 30 June 2020). We considered the data obtained from 2008 to 2017 and made available by the Fire Information for Resource Management System [39,40]. Active fires are identified when the fire reaches a pixel fraction equal to or greater than 0.01% (100 m$^2$) and an average brightness temperature of 800 K.

We also used the burned area product obtained from the Center for Environmental Monitoring and Fire Management, Federal University of Tocantins, which corresponds to the surface affected by fire in the dry season (accessible upon request). The detection of the burned area was with the visual interpretation of the burn scars in two annual images obtained by the Landsat 5 and Landsat 8 satellites (spatial resolutions of 30 m) and from the Resourcesat satellite (spatial resolution of 23.5 m which were resampled to 30 m) [41]. Before the analyses, we revised and recalculated the burned areas for the period 2008–2017 considering surfaces equal or larger than $9 \times 10^{-4}$ km$^2$ (same area as a Landsat pixel).

The annual land use and land cover (LULC) maps from the MapBiomas Project [42], collection 4.1, from 2008 to 2017 (https://mapbiomas.org/en/colecoes-mapbiomas-1?cama_set_language=en, accessed on 12 May 2020), were used to assess the association between fire and type of vegetation affected. These maps are made available at the 30-m spatial resolution. The vegetation types classified by this Project were grouped as fire-sensitive vegetation (forest formations) or fire-resistant vegetation (savanna and grassland formations).

Daily rainfall data from 1 January 2008 to 31 December 2017 were obtained from NASA's Goddard Earth Sciences Data and Information Services Center (3B42 product, version 7), with a spatial resolution of $0.25° \times 0.25°$ (https://gpm.nasa.gov/data/directory, accessed on 21 October 2019). This product is the largest global hydrometeorological database in the world [43]. The 3B42 product combines data from weather stations and GOES-E, GOES-W, Meteosat-5, Meteosat-7, NOAA-12, and Tropical Rainfall Measuring Mission (TRMM) satellite data. Validation studies of these daily rainfall data conducted by [44] yielded satisfactory results for the northern Brazilian region, the region context of the Parque do Araguaia Indigenous Land.

Records of fire combat operations between 2014 and 2017 were also selected from the Integrated Multi-Agency Center for Operational Coordination (Ciman Virtual) (http://queimadas.dgi.inpe.br/queimadas/ciman, accessed on 12 December 2019). This platform was created in 2014 by the National Institute for Space Research (INPE) to make the information obtained by the wildfire operations (costs, number of firefighters, and dates of operation) publicly available [45].

### 2.3. Analysis

Figure 2 shows the flowchart with the main analyzes conducted in this study. The spatial and temporal distribution of rainfall and fire in the study area were analyzed based on maximum values and standard deviation of the burned areas, the number of fire scars, mean monthly rainfall values, duration of the annual dry periods (in days), and the number of dry days before (2008–2012) and during (2013–2017) the creation of FBP, hereafter named BFBP and DFBP, respectively. We considered dry days as with daily precipitation values less than 3 mm of rain and the annual dry season as the largest period of consecutive dry days per year. All burned areas forming distinct polygons were considered different fire scars, regardless of their size or proximity with each other.

Finally, the effects of the dry days and the rainfall (both restricted for the fire season, May to October) on the total burned area and fire cells were analyzed through the Spearman correlations, considering a significance (*p*-value) less than 0.05 ($p < 0.05$).

The spatial distribution of the accumulated number of active fires was analyzed stratifying the study area in a regular grid of $1 \times 1$ km. The active fires data were intersected with the grid. A grid cell with at least one active fire was classified as a "fire cell" regardless of the number of detections within the cell. The other cells were discarded. The total number of annual fire cells during the dry season (May to October) were accumulated monthly and analyzed considering before and during the establishment of FBP.

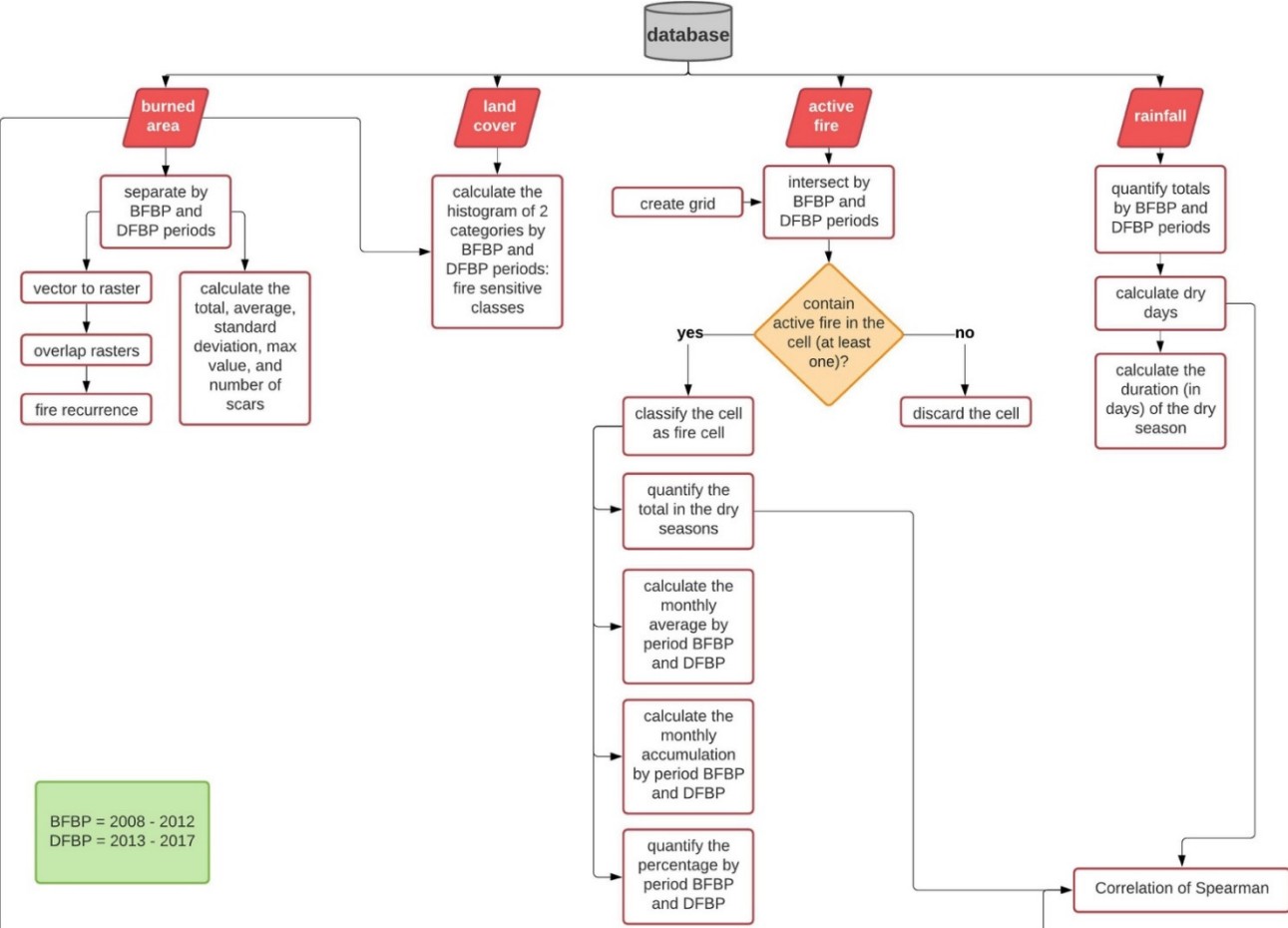

**Figure 2.** Flowchart of the analyses. BFBP = before Fire Brigades Program; DFBP = during Fire Brigades Program.

The relationship between fire and the type of vegetation present in the Parque do Araguaia Indigenous Land was investigated based on the annual percentage of classes and pixel counts before and during the FBP. Then, an analysis was made on an annual basis of vegetation that was impacted by fire relative to the total vegetation resistant to fire (savanna and grass formations) and sensitive to fire (forests).

We calculated the fire recurrence by overlapping the annual totals of burned areas for the BFBP and DFBP periods. Here, the vector-based, annual burned area data were converted into raster format and pixel values were classified in binary values of either 0 (pixels with no burned area) or 1 (pixels with the burned area). We then counted the number of times that a specific pixel presented class 1 before FBP and during FBP. A pixel having class 1 for two times was considered as having one-year recurrence; three times, two-year recurrence; four-times, three-year recurrence; and five-time, four-year recurrence. We also categorized the results according to the type of vegetation impacted (fire-resistant and fire-sensitive).

## 3. Results and Discussion

### 3.1. The Spatial Distribution of Fires in the Parque do Araguaia Indigenous Land

Active fires were found in all months of the BFBP and DFBP periods, except in March, after the implementation of FBP (Figure 3). The highest occurrences were between June and October. The peaks occurred in August (BFBP) and September (BFBP and DFBP), after three months without significant rainfall (Figure 4). In these months, we find traditional fields burned by the indigenous people, along with the activities of the brigades. With the

beginning of the rainy season in October, the number of areas affected by fire gradually reduced until May of the following year.

The fire and rainfall patterns were quite similar before and during the FBP. However, the monthly sums of cells with fires BFBP in May, June, and July were slightly lower (16.1%) than those from DFBP (16.4%). During the peak of the dry season (August and September) and during the transition between dry and wet seasons (October), there was a reduction in the number of cells for the BFBP (83.4%) and DFBP (80.0%) periods. This change in the fire pattern during the FBP is a result of prevention activities conducted by firefighters by using controlled fires and black firebreak [46] and the implementation of Integrated Fire Management in 2015, including prescribed burning at the end of the rainy season and beginning of the dry season [47,48]. Even though the values are not significant, the difference in the patterns between the middle and the end of the dry season during the FBP can be seen as a positive result of the preventive activities of the program.

The smallest and largest burned areas were found in 2009 (1152 km$^2$) and 2010 (11,057 km$^2$) (Table 2), respectively, therefore, BFBP. The lowest number and the highest average size of fire scars were found in 2010. The second-largest burned areas (third in the entire historical series of this study) were found in 2008 (BFBP), while 2011 had the second-lowest value recorded in the period 2008–2017. During the FBP, the lowest total burned area (third lowest value for the entire historical series) was found in 2013, and the highest value (second highest in the historical series) in 2017.

The high values of burned areas in 2010 were influenced by the extreme drought and the duration of the dry season caused by the El Niño climate phenomenon and the warming of the Tropical North Atlantic Ocean current [49–51]. In 2017, there was no extreme climate event, although rainfall was below average in the Cerrado biome from May to October [52] and with the longest drought duration (the longest in the DFBP period). These two events in 2010 and 2017 aid in understanding the impact of rainfall on the size of the burned areas, as pointed out by Mistry [53], Govender et al. [54], and Daldegan et al. [10]. This relationship was also evident in the study area according to the results of Spearman's correlations which had significant, strong, and positive relations between burned areas and dry days burned areas, and rainfall, burned areas and fire cells (Table 3).

The results in Table 2 also refer to an interpretation of landscape fragmentation caused by fire, due to the high standard deviation of the burned area values and the number of fire scars, which do not present a defined pattern before and during the FBP. As the incidence of fire occurs practically every month of the year in the indigenous land (Figure 3), the intra-annual variability of the burned areas can be explained by the quality and quantity of the fuel (biomass) [55].

The longer the time without rainfall, the temperature elevates, and the relative air humidity reduces. This relation affects the flammability of organic fuel and, consequently, the susceptibility to wildfires [56]. Thus, the increase in the extent of the burned areas with an accumulation of organic fuel is facilitated, as this factor also favors the magnitude and spread of the fire at the end of the dry season [57–59]. In this way, the largest extensions of burned areas are more commonly found at the end of the dry season, while smaller ones are more commonly found during the rainy season and at the beginning of the dry season [60].

The average values of fire scars and the maximum values of burned areas were higher, and then lower, respectively, during the FBP, allowing the formation of mosaics from areas with and without fire. This pattern provides a greater environmental heterogeneity, facilitating the regeneration of burned vegetation and avoiding more severe fires at the end of the dry season [61,62]. Despite these positive responses, it is important to condition them to the analysis of the frequency of fire and impacted vegetation.

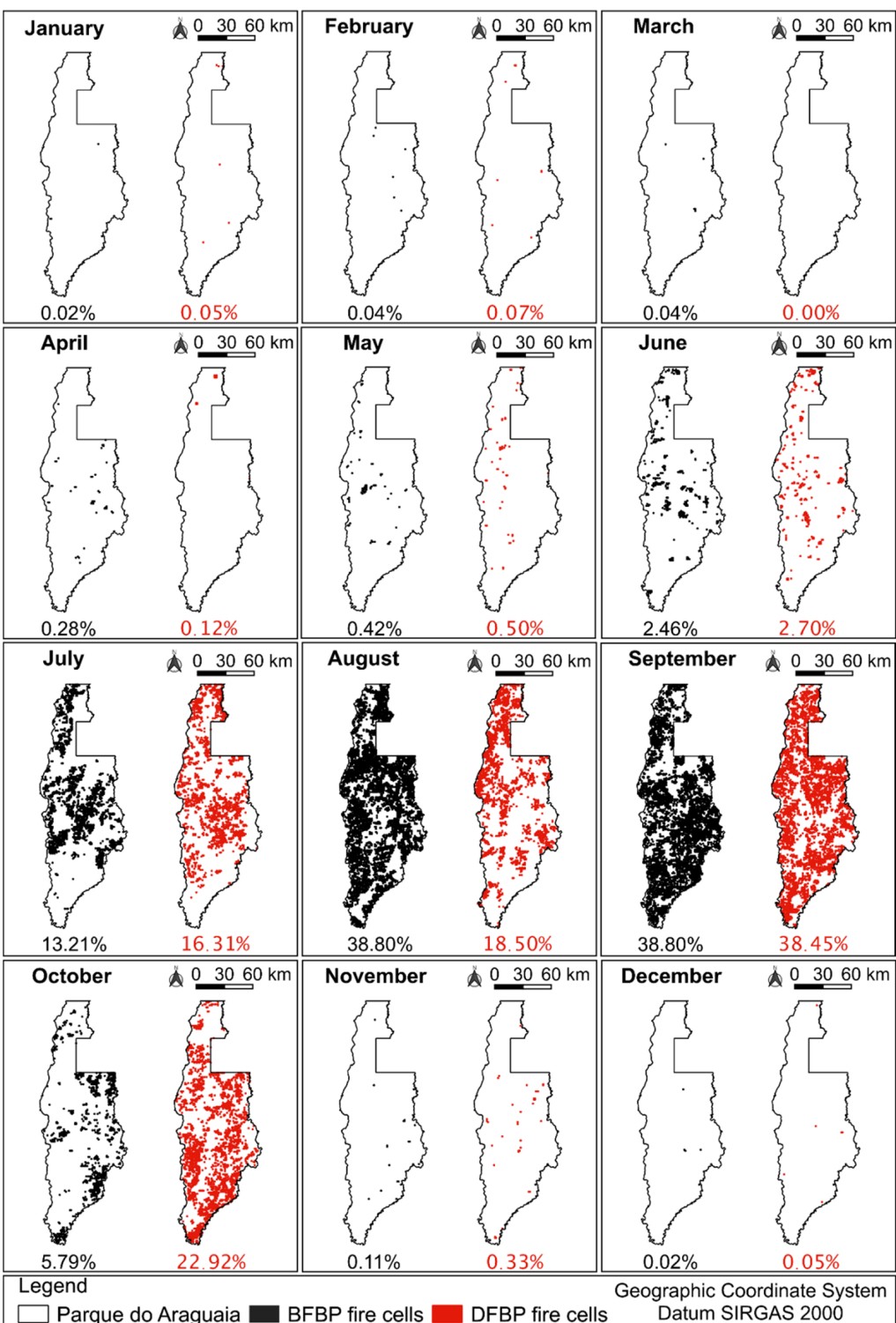

**Figure 3.** Monthly accumulation percentage of fire cells in the Parque do Araguaia Indigenous Land before (2008–2012; black), and during the Federal Brigades Program (2013–2017; red) periods. BFBP = before the Federal Brigades Program; DFBP = during the Federal Brigades Program.

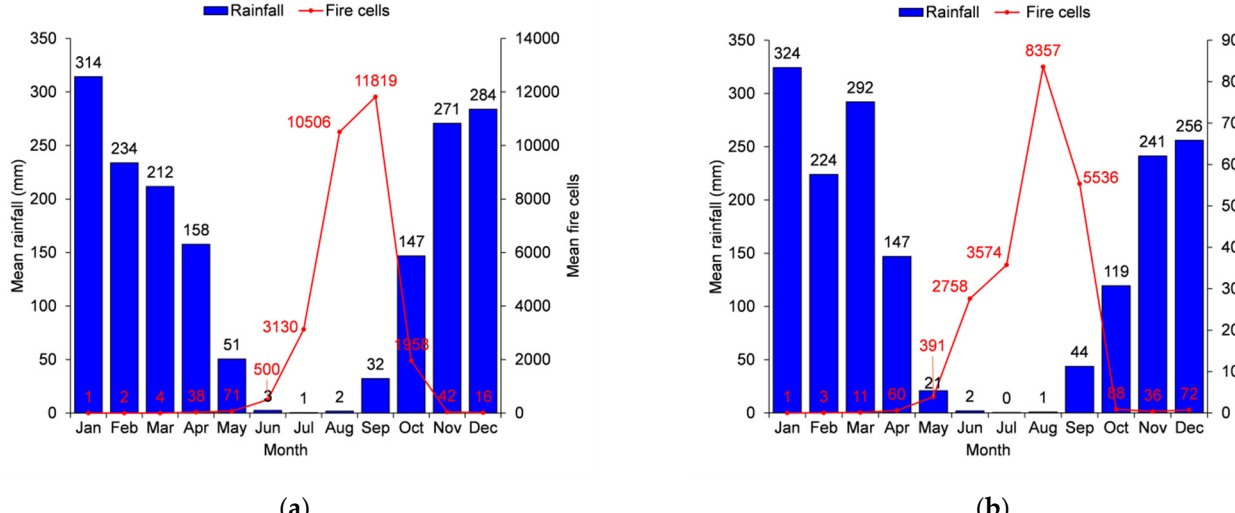

**Figure 4.** Average monthly rainfall and number of fire cells before the Federal Brigades Program (**a**) and during the Federal Brigades Program (**b**).

**Table 2.** Annual description of the burned areas and rainfall in the Parque do Araguaia Indigenous Land, before and during the Federal Brigades Program. BFBP = before the Federal Brigades Program; DFBP = during the Federal Brigades Program.

| Period | Year | Total Burned Area (km²) | Standard Deviation (km²) | Average Burned Area (km²) | Maximum Burned Area (km²) | Total Number of Fire Scars | Total Rainfall (mm) | Duration of the Dry Season (days) |
|---|---|---|---|---|---|---|---|---|
| BFBP | 2008 | 7273 | 321 | 66 | 2508 | 111 | 1837 | 147 |
| | 2009 | 1152 | 16 | 7 | 135 | 172 | 1867 | 118 |
| | 2010 | 11,058 | 1843 | 325 | 10,754 | 34 | 1512 | 170 |
| | 2011 | 1193 | 40 | 9 | 299 | 128 | 1916 | 163 |
| | 2012 | 4072 | 161 | 18 | 2408 | 226 | 1404 | 109 |
| | Mean | 4949 | 476 | 85 | 3221 | 134 | 1707 | 141 |
| DFBP | 2013 | 2819 | 100 | 27 | 744 | 106 | 1891 | 95 |
| | 2014 | 3579 | 88 | 15 | 1116 | 236 | 1788 | 98 |
| | 2015 | 5244 | 167 | 46 | 1220 | 113 | 1404 | 124 |
| | 2016 | 5978 | 273 | 39 | 3262 | 155 | 1498 | 139 |
| | 2017 | 8349 | 332 | 41 | 4242 | 204 | 1780 | 152 |
| | Mean | 5194 | 192 | 34 | 1897 | 163 | 1672 | 122 |

**Table 3.** Spearman correlation results for the whole series (2008–2017) from May to October. Significant values at $p < 0.05$ are depicted in bold.

| Spearman Correlation | Dry Days | Rainfall |
|---|---|---|
| Burned areas | **0.81** | **−0.86** |
| Fire cell | 0.50 | **−0.79** |

### 3.2. Fire Frequency, Areas Affected by Fire, and the Occurrence of Large Forest Fires

The total areas without overlapped burned areas (without incidence of fire and incidence of up to one year) were high in both periods, although, for the other types of fire recurrence, there was a difference between before and during FBP. During FBP, there were a smaller number of areas with one and two years of fire frequency (41% and 11% less during the FBP, respectively). In contrast, areas without fire recurrence and with inter-annual fire recurrence of three and four years were higher during the FBP (16%, 180%, and 751% more before the FBP, respectively) (Table 4). The increase in areas with three and four years of

fire recurrence during the FBP period was high in the natural vegetation, especially in the northern, northeastern, and southeastern portions of the indigenous land (Figure 5).

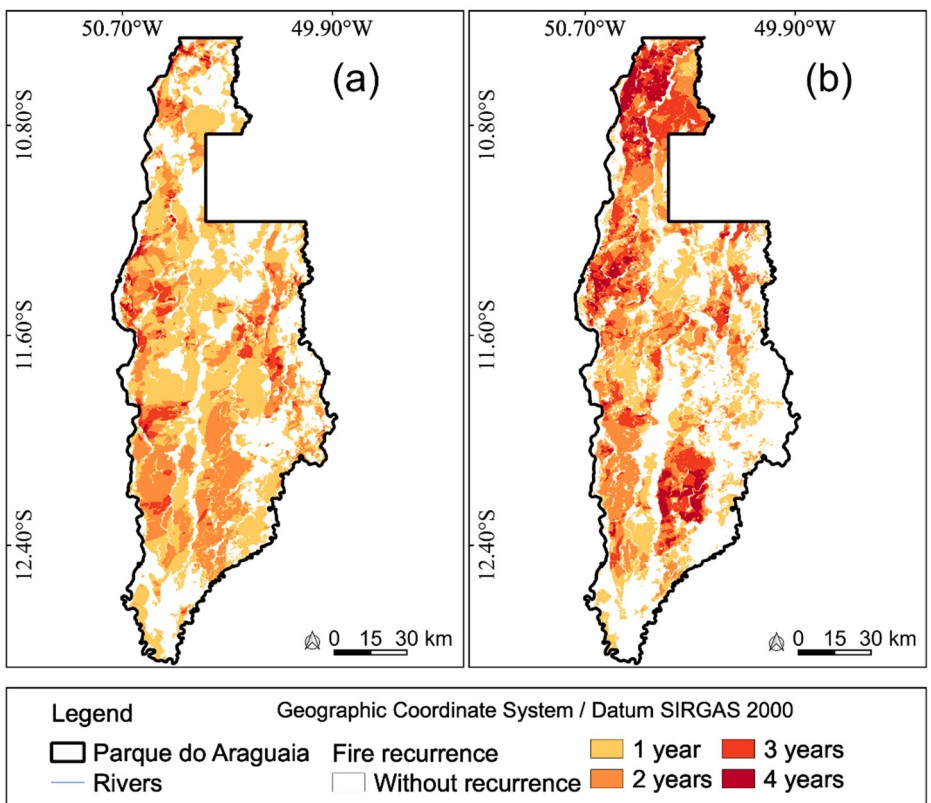

**Figure 5.** Recurrence of burned areas before the Federal Brigades Program (**a**) and during the Federal Brigades Program (**b**).

**Table 4.** Recurrence of burned areas in the Parque do Araguaia Indigenous Land. The differences were calculated using the data for total area of recurrence and area of vegetation types before and during FBP. BFBP = before the Federal Brigades Program; DFBP = during the Federal Brigades Program. Dif. = difference.

| | **Burned Area** | | | **Affected Vegetation in Relation to the Fire Recurrence** | | | | | |
| | | | | **BFBP** | | **DFBP** | | **% Dif.** | |
| **Recurrence** | **BFBP (km²)** | **DFBP (km²)** | **Dif. (%)** | **Sensitive (km²)** | **Fire-Resistant (km²)** | **Sensitive (km²)** | **Fire-Resistant (km²)** | **Sensitive** | **Fire-Resistant** |
|---|---|---|---|---|---|---|---|---|---|
| No. | 5094 | 5903 | 16 | 4838 | 9534 | 1998 | 3435 | −142 | −178 |
| 1 year | 5133 | 3024 | −41 | 295 | 4788 | 125 | 2839 | −137 | −69 |
| 2 years | 3076 | 2733 | −11 | 66 | 3005 | 35 | 2673 | −92 | −12 |
| 3 years | 588 | 1645 | 180 | 9 | 579 | 9 | 1622 | 2 | 64 |
| 4 years | 78 | 664 | 751 | 0 | 78 | 9 | 643 | 95 | 88 |

This recurrence response is related to the annual incidence of fires. Throughout the series (2008–2017), both sensitive vegetation and fire-resistant vegetation were affected by fire (Table 5). The year in which fire-sensitive and fire-resistant vegetation burned the most was in 2010 (before the FBP). With the implementation of the FBP, the mean area of vegetation affected by the fire was smaller in this period, but with progressive growth of affected areas since 2013.

**Table 5.** Percentage of areas affected by the fire and by classes from the MapBiomas Project compared with the total area of each class in the Parque do Araguaia Indigenous Land. BFBP = before the Federal Brigades Program; DFBP = during the Federal Brigades Program.

| Period | Year | Fire-Sensitive Vegetation (%) | Fire-Resistant Vegetation (%) |
|--------|------|-------------------------------|-------------------------------|
| BFBP | 2008 | 18.0 | 62.1 |
| | 2009 | 1.4 | 10.1 |
| | 2010 | 41.6 | 89.9 |
| | 2011 | 2.7 | 10.2 |
| | 2012 | 7.9 | 35.5 |
| Mean | | 14.3 | 41.6 |
| DFBP | 2013 | 2.1 | 24.5 |
| | 2014 | 4.1 | 31.0 |
| | 2015 | 5.0 | 45.9 |
| | 2016 | 8.3 | 52.0 |
| | 2017 | 13.0 | 72.5 |
| Mean | | 6.5 | 45.2 |

The ecological importance of fire in the Cerrado is well-known in the literature [63,64]; nevertheless, the impacts and responses are different in each type of phytophysiognomy found in the biome. Cerrado stricto sensu (grasslands and shrublands) are resistant to fire due to morphology [65]—thick bark, protection of subterranean gems and organs—and plant physiology [66]—translocation of nutrients for underground tissues at the beginning of the drought.

To have a resilient relationship between fire and pyrophytic vegetation, a period of recovery from fire is necessary since each fire overlapping in the same area can increase the damage and mortality of shrub and tree strata, change the plant physiognomy, and delay the passage of specimens to the reproductive stage [67,68].

The need for a fire interval of two or three years to allow regrowth of fire-resistant phytophysiognomies, in some cases, up to four years, is already known [69]. Despite the need for studies to understand the appropriate period of fire recurrence in the ecosystems that exist in the Parque do Araguaia Indigenous Land, with the creation of FBP, the opposite was observed: increase in 3–4-year recurrences and decrease in 1–2-year recurrence.

For fire-sensitive ecosystems, the burned area detection and their recurrence are problematic. In forested areas, there is no positive relationship with the direct incidence of fire, regardless of the period and frequency of occurrence. Forest trees are sensitive to fire [70], and they do not exhibit morphological or physiological characteristics of tree species that occur in the Cerrado stricto sensu. Thus, when there is fire, whether preventive or illegal, there is the possibility of severe impacts on the floristic composition, structure, and survival of individuals [71]. The growth of areas without fire recurrence and with up to one year of burning in an interval of five years during the FBP also indicates another problem—the accumulation of dry organic fuel. This fact provides several impacts, including the homogenization of the landscape, the exclusion of light-dependent herbaceous species, and wildfires [71–73]. Three large wildfires occurred in the indigenous land during the FBP (2014, 2016, and 2017), requesting relatively large costs to the Brazilian public budget. According to data from the Ciman Virtual platform, these fires may have been the result of an accumulation of dry organic fuel.

The environmental impacts of the large wildfires and the high recurrence fires have a great impact on the daily life of traditional peoples as well as on the unique biodiversity of the Parque do Araguaia Indigenous Land. Degradation may also reflect the loss of forested areas (data analysis revealed a 2% reduction in forested areas during the FBP). There is a risk of the continuation of fire degradation as well as the decline of fire-sensitive phytophysiognomies found in the indigenous land if there are no changes in the Program to reduce these detected adversities.

## 4. Conclusions

The FBP is undoubtedly important in the Brazilian environmental policy to reduce wildfires in the Cerrado biome, based on the training and qualification of firefighters to work in protected areas and, whenever necessary, in other areas—federal or state. In the Parque do Araguaia Indigenous Land, the FBP was able to reduce the number of areas affected by the fire at the end of the dry season and helped to create a mosaic of fragments of native vegetation without fire and fire-affected fragments. Negatively, during the execution of the Program in the indigenous land, the number of areas with an accumulation of organic fuel increased and the recurrence of 4–5-year fire also increased.

To mitigate the unfavorable results of the FBP evidenced in this research, it is necessary to incorporate scientific assessments on the impacts of fire on flora and fauna, as well as mapping areas with high recurrence of fire and accumulation of organic fuel before and during the fire prevention and combat activities. Furthermore, monitoring meteorological conditions throughout the year is needed to formulate emergency plans whenever adverse climatic events are identified. In this context, the FBP promotes an opportunity to engage in further discussions with the goal of maintaining and improving the conservation of the savanna ecosystem.

**Author Contributions:** Conceptualization, methodology, project administration, and writing—original draft preparation, A.S.R.d.A.; conceptualization and writing—review and editing, R.M.R.; supervision, data curation and writing—review and editing, E.E.S., R.L., F.L.M.S., J.A.R. and M.G.; review, R.E.d.P.L. and R.R.d.F. All authors have read and agreed to the published version of the manuscript.

**Funding:** This research was supported by the Graduate Program in Geography at the University of Brasília, by the Brazilian Coordination for the Improvement of Higher Education Personnel (CAPES) (Financing code 001), and by the Brazilian National Council for Scientific and Technological Development (CNPq).

**Institutional Review Board Statement:** Not applicable.

**Informed Consent Statement:** Not applicable.

**Data Availability Statement:** All data are public available.

**Acknowledgments:** The authors are grateful for the valuable comments from two anonymous reviewers.

**Conflicts of Interest:** The authors declare no conflict of interest.

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
