# Peer review of "Implementation of Fire Policies in Brazil: An Assessment of Fire Dynamics in Brazilian Savanna"

_sustainability, doi:10.3390/su132011532_

Round 1

Reviewer 1 Report

The major drawback of the submitted manuscript as following:

Abstract is unclear. Please follow the journal guideline as follow (1) Background: Place the question addressed in a broad context and highlight the purpose of the study; (2) Methods: briefly describe the main methods or treatments applied; (3) Results: summarize the article's main findings; (4) Conclusions: indicate the main conclusions or interpretations.

Data and analysis section are not well presented. Detail description is needed.

Database and methods should be separated.

I could not understand that how author(s) produced recurrence of burned areas before FBP (a) and during FBP (b) in figure 4?

Fire incident can cause by human activities or naturally. I am not sure how author (s) distinguish burned area using Aqua and Terra satellite data? Please provide a detail justification in the manuscript.

Most of the fire incident occurred in grassland areas (Figure 1). How author (s) justify the human activities (for example: monocultures) are main cause of burned area?

Please articulate the novelty and the significance of the paper in the abstract, introduction, discussion and conclusion sections.

Please make sure all dash, space, a hyphen, en dash, and capital words would be appropriate throughout the manuscript.

Please make sure the font size in all figures, and text.

Please avoid long sentences and merging the sentences.

Author Response

Dear Editor,

Enclosed, please, find the revised version of the manuscript titled “Implementation of fire policies in Brazil: an assessment of fire dynamics in Brazilian Savanna”. We did our best trying to incorporate all suggestions from the reviewers. Below, you can find the response to the comments from the reviewers point-by-point. Please see the attachment.

  1. Abstract is unclear. Please follow the journal guideline as follow (1) Background: Place the question addressed in a broad context and highlight the purpose of the study; (2) Methods: briefly describe the main methods or treatments applied; (3) Results: summarize the article's main findings; (4) Conclusions: indicate the main conclusions or interpretations.

Thanks a lot for the comment. We rewrote the Abstract presenting a brief introduction of the subject, followed by the statement of the objective, description of the major data sets, and the main results and conclusions.

  1. Data and analysis section are not well presented. Detail description is needed.

Thanks for this important comment. We added a flowchart at the beginning of the 2.3 Analysis section to facilitate the comprehension of methodological approach proposed in this study. All paragraphs of the 2.2. Data sets and 2.3. Analysis were rewritten for clarity purpose.

  1. Database and methods should be separated.

In the Materials and Methods section, we divided it into Study Area, Data Sets, and Analysis for better clarity.

  1. I could not understand that how author(s) produced recurrence of burned areas before FBP (a) and during FBP (b) in Figure 4?

For clarity, we rewrote the second last paragraph of the section 2.3. Analysis as:

We calculated the fire recurrence by overlapping the annual totals of burned areas for the BFBP and DFBP periods. Here, the vector-based, annual burned area data were converted into raster format and pixel values were classified in binary values of either 0 (pixels with no burned area) or 1 (pixels with the burned area). We then counted the number of times that a specific pixel presented class 1 before FBP and during FBP. A pixel having class 1 for two times was considered as having one-year recurrence; three times, two-year recurrence; four times, three-year recurrence; and five time, four-year recurrence. We also categorized the results according to the type of vegetation impacted (fire-resistant and fire-sensitive).

  1. Fire incident can cause by human activities or naturally. I am not sure how author (s) distinguish burned area using Aqua and Terra satellite data? Please provide a detail justification in the manuscript.

In fact, we did not interpret any Terra MODIS and Aqua MODIS satellite images. The MCD14ML, collection 6, active fire products we downloaded from the USGS website are derived from these two satellites.

  1. Most of the fire incident occurred in grassland areas (Figure 1). How author (s) justify the human activities (for example, monocultures) are the main cause of burned area?

In the Cerrado biome, most of the fires are related to the agricultural activities, but this is not true for the Parque do Araguaia Indigenous Land. Since this land is protected, we do not find agriculture there. Then, fires in this indigenous land are related to traditional cultivation methods or are illegally set by hunters and land grabbers.

  1. Please, articulate the novelty and the significance of the paper in the abstract, introduction, discussion and conclusion sections.

Thanks a lot for this important comment. In the Abstract, we added the following sentence: “To our best knowledge, there is no study analyzing the effectiveness of the FBP-based actions to prevent wildfires in Brazil”. At the end of the Introduction section, we added the following sentence:

The research question of this study is: How effective is the FBP in reducing the size, number, and recurrence of fires in the Brazilian Cerrado? To our best knowledge, this is the first study to present an analysis of effectiveness of the FBS-based actions to prevent wildfires in the Brazilian Cerrado.

  1. Please, make sure all dash, space, a hyphen, en dash, and capital words are appropriate throughout the manuscript.

Thanks for the suggestion. We tried to correct all existing “typos” in the revised manuscript.

  1. Please, make sure the font size in all figures, and text.

We made our best effort to standardize the font size of the figures.

  1. Please, avoid long sentences and merging the sentences.

We checked and tried to correct them.

Reviewer 2 Report

This study presents an interesting assessment of fire prevention policies. While the current manuscript requires some improvements. In addition, there are substantial grammar and tense errors, please revise your writing.

Comments:

  1. Line 31-32: obtained in this study indicate – suggest
  2. Line 35: All keywords should have the same format. environmental policy – fire policy
  3. Line 37-49: The introduction is insufficient and needs to be improved. The first paragraph is redundant. It is better to start the introduction with a big picture. You can briefly introduce the fire impacts on society (Viegas et al., 2009), urban development (Moritz & Stephens, 2008), water quality (Rhoades et al., 2019), and carbon cycle (Wei et al., 2018). Then, introduce policies related to fire monitoring, management, and prevention. Because your article has a lot of content related to fire prevention policies, please review some similar policies. Therefore, you can move to your study area in the second paragraph.
  4. Line 50-71: Please make these paragraphs shorter. You may merge them. A brief introduction of the policy is enough. In addition, you need to add more information related to the policy implementation.  
  5. 2 million km² - 2×106 km2, please revise this problem throughout the manuscript.
  6. Line 91-95: What is the purpose of this paragraph? If this is a description of your study area, please move it to method or delete it.
  7. At the end of the introduction, please address your scientific questions and hypotheses (optional).
  8. Line 107-110: delete this sentence, it is no use. You can merge this paragraph with the next one.
  9. Line 123: delete has
  10. Line 140: What is 800 oK?
  11. Please provide the website link of each data set used in this project.
  12. Line 225-226: The fire statistics are too simple. Some fire indices can be used to describe fires such as fire cycle (Wei & Larsen, 2018). For example, the fire cycle can represent the fire size and frequency. Giglio (2007) presented the fire cycle calculation with remote sensing data.
  13. Fire severity is also important. Is the fire severity data available? If available, please include it.
  14. I am wondering about the fire scare. Does a scar represent one fire?

Giglio, L. (2007). Characterization of the tropical diurnal fire cycle using VIRS and MODIS observations. Remote Sensing of Environment, 108(4), 407-421.

Moritz, M. A., & Stephens, S. L. (2008). Fire and sustainability: considerations for California’s altered future climate. Climatic Change, 87(1), 265-271.

Rhoades, C. C., Chow, A. T., Covino, T. P., Fegel, T. S., Pierson, D. N., & Rhea, A. E. (2019). The legacy of a severe wildfire on stream nitrogen and carbon in headwater catchments. Ecosystems, 22(3), 643-657.

Viegas, D. X., Ribeiro, L., Viegas, M., Pita, L., & Rossa, C. (2009). Impacts of fire on society: Extreme fire propagation issues. In Earth observation of wildland fires in Mediterranean ecosystems (pp. 97-109): Springer.

Wei, X., Hayes, D. J., Fraver, S., & Chen, G. (2018). Global Pyrogenic Carbon Production During Recent Decades Has Created the Potential for a Large, Long‐Term Sink of Atmospheric CO2. Journal of Geophysical Research: Biogeosciences, 123(12), 3682-3696.

Wei, X., & Larsen, C. (2018). Assessing the Minimum Number of Time Since Last Fire Sample-Points Required to Estimate the Fire Cycle: Influences of Fire Rotation Length and Study Area Scale. Forests, 9(11), 708. https://www.mdpi.com/1999-4907/9/11/708

Author Response

Dear Editor,

Enclosed, please, find the revised version of the manuscript titled “Implementation of fire policies in Brazil: an assessment of fire dynamics in Brazilian Savanna”. We did our best trying to incorporate all suggestions from the reviewers. Below, you can find the response to the comments from the reviewers point-by-point. Please see the attachment.

  1. Line 31-32: obtained in this study indicate – suggest.

Changed, as suggested.

  1. Line 35: All keywords should have the same format. environmental policy – fire policy.

Changed, as suggested.

  1. Line 37-49: The introduction is insufficient and needs to be improved. The first paragraph is redundant. It is better to start the introduction with a big picture. You can briefly introduce the fire impacts on society (Viegas et al., 2009), urban development (Moritz & Stephens, 2008), water quality (Rhoades et al., 2019), and carbon cycle (Wei et al., 2018). Then, introduce policies related to fire monitoring, management, and prevention. Because your article has a lot of content related to fire prevention policies, please review some similar policies. Therefore, you can move to your study area in the second paragraph.

Thanks for such an important comment. The beginning of the Introduction was rewritten as follow:

Wildfires disturb natural environments, affect socio-economic activities, and put the health and the life of people in danger [1]. More specifically, wildfires consume most of surface organic layers and reduce stream water quality by increasing the inputs of soil erosional sediments and fertilizers [2]. Fires causes landscape fragmentation [3], loss of biodiversity [4], and release particulates carbon stored in biomass and soils into the atmosphere, mostly in the form of CO2, but also as CO, CH4, and CH3Cl [5-7]. Whenever dry season is prolonged, fire severity and intensity can be aggravated due to the increase of the dry organic fuels that can cause extensive and uncontrolled wildfires [8-10]. Moritz and Stephens [11] recommended that future cities and rural communities should be created considering low susceptibility to wildfires.

In Brazil, fire in rural areas is mainly associated with public and illegal land cleaning for land speculation, land preparation for crop cultivation, pasture renewal, and charcoal production [12]. Fire prevention policy in Brazil is mainly based on field environmental law enforcement operations to combat and control illegal fire activities [13,14]. In addition, to reduce the negative consequences of fires and to comply with the goals established in international agreements, Brazil has regulated rules on the use of fire [15], created specific databases and public policies to support fire monitoring activities [16,17], and fostered projects based on technical cooperation agreements with other countries [18,19]. The Brazilian Forest Code, approved in 2012 (Law no. 12,651 of 25 May 2012), allows the use of fire management for conservation purposes in private and public protected areas, except in traditional communities.

  1. Line 50-71: Please make these paragraphs shorter. You may merge them. A brief introduction of the policy is enough. In addition, you need to add more information related to the policy implementation.

The two paragraphs were merged into one and made shorter, according to the reviewer suggestion. The revised paragraph is as follow:

Brigades are also another important means to carry out fire prevention activities, environmental education, and to combat wildfires in Brazil [20,21]. The FBP was implemented in 2012 in Brazil as part of the Sectorial Health Plan for the Mitigation and Adaption to Climate Change [22]. The objective of the FBP is to train and hire firefighters to reduce the extent and the number of fire occurrences during the dry season, combat wildfires, and carry out activities to prevent uncontrolled fires [23]. Brigades are hired whenever the Ministry of Environment (MMA) declares year-based emergency periods during the dry seasons in different regions of Brazil [24].

  1. 2 million km² - 2×106 km2, please revise this problem throughout the manuscript.

Thanks for the suggestion. The terms “thousand” and “million” were changed to (× 103) and (× 106) throughout the text.

  1. Line 91-95: What is the purpose of this paragraph? If this is a description of your study area, please move it to method or delete it.

This paragraph was moved to the 2.1 Study Area section.

  1. At the end of the introduction, please address your scientific questions and hypotheses (optional).

The following sentences were added in the end of the Introduction section:

The research question of this study is: How effective is the FBP in reducing the size, number, and recurrence of fires in the Brazilian Cerrado? To our best knowledge, this is the first study to present an analysis of effectiveness of the FBS-based actions to prevent wildfires in the Brazilian Cerrado.

  1. Line 107-110: delete this sentence, it is no use. You can merge this paragraph with the next one.

The two paragraphs were merged and much shortened, as suggested by the reviewer. The new paragraph became:

The study area corresponded to the Parque do Araguaia Indigenous Land, which has an area of approximately 13 ´ 103 km². This indigenous land corresponds to a large seasonal wetland and the most important intracratonic Quaternary sedimentary basin of South America, flooded mainly by rainfall and ground water [35,36]. The annual flooding occurs during the rainy season mostly from the Javaés and Araguaia rivers [35]. The region is characterized by a mosaic of phytophysiognomies that are either resistant to fire (shrublands and wooded grasslands) or sensitive to fire (forestlands) [37]. The predominant climate is tropical, with an annual average temperature ranging from 22 °C to 26 °C [38]. The dry season occurs between May and September while the rainy season occurs between October and April.

  1. Line 123: delete has.

The term “Since 2013, IBAMA has considered…” was changed to “Since 2013, IBAMA considers…”.

  1. Line 140: What is 800 oK?

Sorry. This typo was correct for 800 K (800 degree Kelvin).

  1. Please provide the website link of each data set used in this project.

We added the links for downloading the following data sets in the revised version of the manuscript: MODIS MCD14ML; MapBiomas Project, NASA´s TRMM; and Ciman Virtual. Burned area product from the Center for Environmental Monitoring and Fire Management is not available for download, but can be accessed upon request.

  1. Line 225-226: The fire statistics are too simple. Some fire indices can be used to describe fires such as fire cycle (Wei & Larsen, 2018). For example, the fire cycle can represent the fire size and frequency. Giglio (2007) presented the fire cycle calculation with remote sensing data.

We believe the use of fire cycle approach proposed, for example, by these two authors demand for long time series of data for accurate analysis. For example, Wei & Larsen (2018) used historical fire and weather data from 1969 to 2012. Giglio (2007) used daily fire products from1997 to 2005. Although we used 10-year data set, we preferred to slipt them into two sets in order to analyze the effectiveness of implementation of the Fire Brigades Program in Brazil.

  1. Fire severity is also important. Is the fire severity data available? If available, please include it.

Unfortunately, the MODIS products do not quality fires in terms of severity neither the other fire data we used in this study.

  1. I am wondering about the fire scar. Does a scar represent one fire?

All burned areas forming distinct polygons were considered different fire scars, regardless of its size or proximity with each other. This explanation was added in the revised version of the manuscript.

Giglio, L. (2007). Characterization of the tropical diurnal fire cycle using VIRS and MODIS observations. Remote Sensing of Environment, 108(4), 407-421.

Moritz, M. A., & Stephens, S. L. (2008). Fire and sustainability: considerations for California’s altered future climate. Climatic Change, 87(1), 265-271.

Rhoades, C. C., Chow, A. T., Covino, T. P., Fegel, T. S., Pierson, D. N., & Rhea, A. E. (2019). The legacy of a severe wildfire on stream nitrogen and carbon in headwater catchments. Ecosystems, 22(3), 643-657.

Viegas, D. X., Ribeiro, L., Viegas, M., Pita, L., & Rossa, C. (2009). Impacts of fire on society: Extreme fire propagation issues. In Earth observation of wildland fires in Mediterranean ecosystems (pp. 97-109): Springer.

Wei, X., Hayes, D. J., Fraver, S., & Chen, G. (2018). Global Pyrogenic Carbon Production During Recent Decades Has Created the Potential for a Large, Long‐Term Sink of Atmospheric CO2. Journal of Geophysical Research: Biogeosciences, 123(12), 3682-3696.

Wei, X., & Larsen, C. (2018). Assessing the Minimum Number of Time Since Last Fire Sample-Points Required to Estimate the Fire Cycle: Influences of Fire Rotation Length and Study Area Scale. Forests, 9(11), 708. https://www.mdpi.com/1999-4907/9/11/708

Round 2

Reviewer 1 Report

I think, author (s) provided all comments and revised the manuscript suggested by reviewer (s). Thus, manuscript can be accepted. 

Author Response

Many thanks for accepting our revisions and for taking your time reading our research.

Reviewer 2 Report

The manuscript has been well revised, but it still has some unsolved questions. In addition, please correct your grammar and tense errors. I did not figure out everyone. please double-check that.

  1. Line 23-24: Please rewrite “Fire occurrences in Brazil are highest in the Cerrado (tropical savanna) biome. Recently, human activities and climate change have intensified fire occurrences in this biome.”.
  2. Line 27-35: Please reduce your descriptions of these datasets and briefly address your research method. In addition, please give more descriptions of your results. It is too simple.
  3. The introduction should have a brief review of existing similar studies. Such as similar policies applied in other regions. I do not mean you need to talk more in Brazil. I saw the added paragraph. But it is not a review. It is necessary to highlight why your study is required? What are the innovations of your study? A review can help you to comprehensively answer these two questions.
  4. Line 73-80: Delete this paragraph. This is not a part of the introduction. The precipitation information can be moved to the method.
  5. Line 115: areas – area
  6. Line 121 – 129: Correct your writing.
  7. Line 130: from – obtained from
  8. Line 137: approximate area of Landsat pixels – same area as a Landsat pixel
  9. Line 178: 1 km × 1 km – 1 × 1 km
  10. Line 185: was investigated considering – Wrong English.
  11. Line 232: The second highest number of burned areas – The second-largest burned area

Author Response

RESPONSES TO THE COMMENTS FROM THE REVIEWER # 2

  1. The manuscript has been well revised, but it still has some unsolved questions. In addition, please correct your grammar and tense errors. I did not figure out everyone. Please double-check that.

Thanks for taking your time revising our study and providing such important comments. We did our best to correct English writing and to improve the second version of the manuscript based on your new comments.

  1. Line 23-24: Please rewrite “Fire occurrences in Brazil are highest in the Cerrado (tropical savanna) biome. Recently, human activities and climate change have intensified fire occurrences in this biome.”

Thanks for the suggestion. We removed this sentence in order to fit the Abstract into the 200-word size.

  1. Line 27-35: Please reduce your descriptions of these datasets and briefly address your research method. In addition, please give more descriptions of your results. It is too simple.

Thanks a lot for the suggestion. The Abstract was rewritten as:

In 2002, the Brazilian government implemented the Federal Brigades Program (FBP), a fire policy strategy to hire and train firefighters to combat wildfires. This study analyzed the impact of this program on fire behavior before (2008-2012) and after (2013-2017) its implementation in the Parque do Araguaia Indigenous Land, the largest indigenous territory with the highest occurrence of fires in the Brazilian tropical savanna. We analyzed the annual pattern of fire incidence in the dry season, the fire impact per vegetation type, the recurrence, and the relationship between fire and precipitation. The data sets were based on active fire products derived from the Moderate Resolution Imaging Spectroradiometer (MODIS), the Landsat and Resourcesat-based burned area products, and the records of the fire combat operations. Our results showed that FBP contributed to the reduction of the number of areas affected by fires and to the formation of a more heterogeneous environment composed of fire-resistant and fire-sensitive native vegetation fragments. On the other hand, after the implementation of the FBP, there was an increase in the recurrence of 3-4 years of fires. We concluded that the FBP is an important public policy capable of providing improvements in fire management activities.

  1. The introduction should have a brief review of existing similar studies. Such as similar policies applied in other regions. I do not mean you need to talk more in Brazil. I saw the added paragraph. But it is not a review. It is necessary to highlight why your study is required? What are the innovations of your study? A review can help you to comprehensively answer these two questions.

Thanks a lot for raising this point one more time. We added two new paragraphs in the end of the Introduction section:

Most of the scientific publications related to the rural fire brigades deal with safety, health, and psychological issues [e.g., 26-28]. There are few studies investigating the effectiveness of rural fire brigades hired to reduce burned areas. An exception is the recent research conducted by Oliveira et al. [29] reported a reduction of 12%, on average, of burned areas in conservation units located in the Brazilian Cerrado (tropical savanna) with brigades, compared with those units without brigades. The same study found a reduction of additional 6% of burned areas in the Cerrado´s conservation units with both fire suppression and prevention activities in comparison with those units with only suppression practices.

After decades of frustrate attempts of implementing zero-fire policies, Brazil started to implement fire management policies, reintroducing fire as a management tool in fire-prone ecosystems, which is the case of the Brazilian Cerrado [30,31]. Firefighters hired by the FBP are important players of these policies. They can act not only in federal conservation units of integral protection or sustainable use but also in indigenous lands, public forests, federal rural settlements, and quilombos. The research question of this study is how effective is the FBP in reducing the size, number, and recurrence of fires in the Brazilian Cerrado? To our best knowledge, there is lack of studies regarding the effectiveness of the FBP-based actions to prevent wildfires in Brazil. To evaluate these possible effects of FBP, this study aimed to compare the spatial variability of fires before (2008-2012) and during (2013-2017) the implementation of the FBP in the Parque do Araguaia Indigenous Land.

  1. Line 73-80: Delete this paragraph. This is not a part of the introduction. The precipitation information can be moved to the method.

The two paragraphs regarding the Cerrado biome were excluded, following the reviewers recommendation.

  1. Line 115: areas – area

Corrected.

  1. Line 121 – 129: Correct your writing.

Thanks for the suggestion. We rewrote as follow:

We selected the Moderate Resolution Imaging Spectroradiometer (MODIS) Active Fire product (MCD14ML), collection 6, which is derived applying thresholds to the brightness temperatures from the middle-infrared and thermal infrared spectral bands of the MODIS sensor onboard the Aqua and Terra platforms (https://firms.modaps.eosdis.nasa.gov/ download/). We considered the data obtained from 2008 to 2017 and made available by the Fire Information for Resource Management System [52,53]. Active fires are identified when the fire reaches a pixel fraction equal to or greater than 0.01% (100 m²) and an average brightness temperature of 800 K.

  1. Line 130: from – obtained from

Corrected, as suggested.

  1. Line 137: approximate area of Landsat pixels – same area as a Landsat pixel

Corrected, as suggested.

  1. Line 178: 1 km × 1 km – 1 × 1 km

Corrected, as suggested.

  1. Line 185: was investigated considering – Wrong English.

Corrected, as suggested.

  1. Line 232: The second highest number of burned areas – The second-largest burned area

Corrected, as suggested.

Round 3

Reviewer 2 Report

The current manuscript is ready to be published.